# LEARNING TO GROW PRETRAINED MODELS FOR EFFICIENT TRANSFORMER TRAINING

**Peihao Wang**[1]* **Rameswar Panda**[2] **Lucas Torroba Hennigen**[4] **Philip Greengard**[3]
**Leonid Karlinsky**[2] **Rogerio Feris**[2] **David D. Cox**[2] **Zhangyang Wang**[1] **Yoon Kim**[4]

[1]University of Texas at Austin, [2]MIT-IBM Watson AI Lab, [3]Columbia University, [4]MIT
{peihaowang,atlaswang}@utexas.edu, {rpanda, leonidka, david.d.cox}@ibm.com,
rsferis@us.ibm.com, pg2118@columbia.edu, {lucastor, yoonkim}@mit.edu

## ABSTRACT

Scaling transformers has led to significant breakthroughs in many domains, leading to a paradigm in which larger versions of existing models are trained and released on a periodic basis. New instances of such models are typically trained completely from scratch, despite the fact that they are often just scaled-up versions of their smaller counterparts. How can we use the implicit knowledge in the parameters of smaller, extant models to enable faster training of newer, larger models? This paper describes an approach for accelerating transformer training by learning to grow pretrained transformers, where we learn to linearly map the parameters of the smaller model to initialize the larger model. For tractable learning, we factorize the linear transformation as a composition of (linear) width- and depth-growth operators, and further employ a Kronecker factorization of these growth operators to encode architectural knowledge. Extensive experiments across both language and vision transformers demonstrate that our learned Linear Growth Operator (LiGO) can save up to 50% computational cost of training from scratch, while also consistently outperforming strong baselines that also reuse smaller pretrained models to initialize larger models.[1]

## 1 INTRODUCTION

The transformer architecture (Vaswani et al., 2017) has emerged as a general purpose architecture for modeling many structured domains (Devlin et al., 2019; Brown et al., 2020; Rives et al., 2021; Dosovitskiy et al., 2021; Touvron et al., 2021a). Perhaps more so than other architectures, the transformer empirically seems to have inductive biases that make it especially amenable to scaling (Rosenfeld et al., 2019; Kaplan et al., 2020), which has led to a paradigm in which larger versions of smaller, existing models are trained and released on a periodic basis (e.g., the GPT lineage of models (Radford et al., 2018; 2019; Brown et al., 2020)). New instances of such models are typically trained completely from scratch, despite the fact that they are often scaled-up versions of their smaller counterparts. Given the compute required to train even the smaller models, we argue that training each model from scratch is wasteful, and that prior knowledge implicit in the parameters of smaller pretrained models should be leveraged to enable faster training of larger models.

One approach to this problem is through the lens of *model growth*, wherein a smaller model's pretrained parameters are used to initialize a subset of the larger model's parameters. While earlier works generally froze the parameters initialized from the pretrained model and only trained the new (randomly initialized) parameters (Fahlman & Lebiere, 1989; Fahlman, 1990; Gutstein et al., 2008), subsequent work has shown that copying a subset of the pretrained parameters to initialize the new parameters and then finetuning the entire network significantly accelerates training and sometimes even leads to better performance (Chen et al., 2015). When applied to modern transformers, these mechanisms roughly translate to a depth-expansion operator in which pretrained models are stacked (or combined with identity layers) to initialize deeper transformers (Gong et al., 2019; Yang et al., 2020), and a width-expansion operator in which the smaller model's matrices are copied to initialize the larger model's matrices (e.g., in block-diagonal fashion) (Chen et al., 2021; Gu et al., 2020).

---

*Work done during an internship at MIT-IBM Watson AI Lab.
[1]Project page: `https://vita-group.github.io/LiGO/`

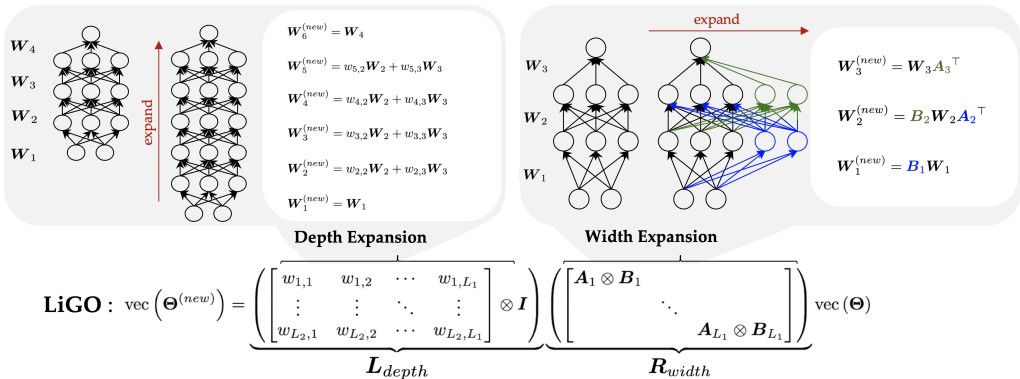

$$\textbf{LiGO}: \ \text{vec}\left(\boldsymbol{\Theta}^{(new)}\right) = \underbrace{\left(\begin{bmatrix} w_{1,1} & w_{1,2} & \cdots & w_{1,L_1} \\ \vdots & \vdots & \ddots & \vdots \\ w_{L_2,1} & w_{L_2,2} & \cdots & w_{L_2,L_1} \end{bmatrix} \otimes \boldsymbol{I}\right)}_{\boldsymbol{L}_{depth}} \underbrace{\left(\begin{bmatrix} \boldsymbol{A}_1 \otimes \boldsymbol{B}_1 & & \\ & \ddots & \\ & & \boldsymbol{A}_{L_1} \otimes \boldsymbol{B}_{L_1} \end{bmatrix}\right)}_{\boldsymbol{R}_{width}} \text{vec}\left(\boldsymbol{\Theta}\right)$$

**Figure 1:** Our linear growth operator (LiGO) accelerates training by using the weights of a smaller model $\boldsymbol{\Theta}$ to initialize the weights of the larger model $\boldsymbol{\Theta}^{(new)}$. LiGO is parameterized as a sparse linear map $\boldsymbol{M}$ that can be decomposed into width- and depth-expansion operators. The width-operator $\boldsymbol{R}_{width}$ and depth-operator $\boldsymbol{L}_{depth}$ are structured matrices obtained from Kronecker products of smaller matrices which encode architectural knowledge by grouping parameters into layers and neurons. While we show the expansion operators for simple multi-layer perceptrons for illustrative purposes, in practice we apply LiGO to enable faster training of transformer networks. In our approach, we learn the growth matrix $\boldsymbol{M}$ with a 100 steps of SGD, use this to initialize the larger model, and then continue training as usual. Best viewed in color.

Noting the empirical effectiveness of such recipes, we observe that existing mechanisms generally do not have a learning component (e.g., randomly copying over neurons for width-expansion or stacking consecutive layers for depth-expansion). This paper instead proposes an efficient, data-driven approach for *learning to grow* transformers. In particular, our approach frames the problem of initializing the larger model's parameters as learning a linear mapping from the smaller model's parameters, i.e., $\boldsymbol{\Theta}^{(large)} = \boldsymbol{M}\boldsymbol{\Theta}^{(small)}$ where $\boldsymbol{\Theta}^{(small)}$ and $\boldsymbol{\Theta}^{(large)}$ are the vectorized parameters of the small/large models. Due to the high dimensionality of the parameters, this mapping is completely intractable to learn without any restrictions on $\boldsymbol{M}$. We thus factorize the linear mapping to be a composition of sparse width- and depth-expansion operators, $\boldsymbol{M} = \boldsymbol{L}_{depth}\boldsymbol{R}_{width}$, where both width and depth matrices are further factorized to be a Kronecker product of smaller matrices that express architectural knowledge (e.g., through grouping parameters by layers and neurons). We show that our growth operators can represent existing approaches such as layer-stacking and neuron-copying as special cases. We find that with a small amount of learning on $\boldsymbol{M}$ (e.g., 100 gradient steps) to initialize the larger model, we can significantly accelerate training of both vision and language transformers. Figure 1 illustrates our approach.

We apply our learned linear growth operator (LiGO) to popular families of models—BERT (Devlin et al., 2019), RoBERTa (Liu et al., 2019), GPT2 (Radford et al., 2019), and ViT (Dosovitskiy et al., 2021; Touvron et al., 2021a;b)—and find that LiGO can consistently improve transformer training efficiency over the traditional way of training from scratch across domains and model sizes. For instance, LiGO saves $44.7\%$ and $22.5\%$ FLOPs for training BERT-Base and GPT2-Medium from scratch by reusing pretrained smaller models that are half as big. Similarly, for vision transformers, when using DeiT-S (Touvron et al., 2021a) for initialization, LiGO yields $55\%$ savings in FLOPs with no performance drop on ImageNet (Deng et al., 2009). These FLOPs savings directly translate to similar wall clock savings. We further find that models trained using LiGO achieve similar performance to the trained-from-scratch baselines when transferred to downstream tasks.

## 2 RELATED WORK

**Efficient training.** Efficient training of transformers has been studied from multiple perspectives. Some methods that are orthogonal to our work include mixed precision training (Shoeybi et al., 2019), large batch optimization (You et al., 2019), distributed training (Huang et al., 2019), and dropping layers (Zhang & He, 2020) or tokens (Hou et al., 2022). Knowledge inheritance (Qin et al., 2021) explores knowledge distillation during pretraining to efficiently learn larger transformers. Progressive training, which first trains a small transformer with few layers and then gradually expands by stacking layers, has also been applied to accelerate transformer training (Gong et al., 2019; Yang et al., 2020; Li et al., 2022; Shen et al., 2022). Net2Net Chen et al. (2015) uses function-preserving transformations to grow width by copying neurons and depth by using identity layers. Recently, bert2BERT (Chen et al., 2021) extends Net2Net to transformers. In contrast to these approaches, our approach learns to (linearly) transform the parameters of a smaller model to initialize a

larger model. While there is a line of work on learning to grow neural networks in a data-driven way, these methods are in general difficult to apply to modern-scale transformers since they (for example) involve growing a single neuron at a time or employ expensive optimization/search procedures (Wei et al., 2016; Cai et al., 2018; Wu et al., 2019; 2021; Evci et al., 2022).

**Network initialization.** Our work is also related to work on neural network initialization. Existing works include controlling the norm of the parameters (Mishkin & Matas, 2015; Kilcher et al., 2018; Dai et al., 2019; Wu et al., 2019; Glorot & Bengio, 2010) or replacing the normalization layers (Brock et al., 2021; Zhang et al., 2019; Huang et al., 2020). MetaInit (Dauphin & Schoenholz, 2019) proposes an automatic method that optimizes the norms of weight tensors to minimize the gradient quotient on minibatches of random Gaussian samples. GradInit (Zhu et al., 2021) learns to initialize larger networks by adjusting norm of each layer. Our work focuses on using smaller pretrained transformers to better initialize larger transformers, which remains an understudied problem.

**Structured matrices.** Finally, our work is also related to structured matrices which are typically used to replace dense weight matrices for reducing training and inference computation cost. Examples include sparse and low rank matrices (Chiu et al., 2021; Han et al., 2015), Chebyshev matrices (Tang et al., 2019), Toeplitz matrices (Sindhwani et al., 2015), Kronecker-product matrices (Zhang et al., 2015), and butterfly matrices (Dao et al., 2019). A unified framework to learn a broad family of structured matrices is presented in Sindhwani et al. (2015). Dao et al. (2022) propose Monarch matrices, which inherit the expressiveness of butterfly matrices and achieve reasonable accuracy-efficiency tradeoffs in many applications. While our approach is inspired by these works, we propose to grow pretrained models by learning structured sparse linear operators with Kronecker factorization, which to our knowledge has not been explored in the literature.

## 3 PROPOSED APPROACH

**Notation.** We denote the parameters of a neural network with $L$ layers and $D$ dimensions as $\boldsymbol{\Theta}_{L,D} = [\boldsymbol{W}_1 \;\; \cdots \;\; \boldsymbol{W}_L]^\top \in \mathbb{R}^{LD \times D}$, where $\boldsymbol{W}_l \in \mathbb{R}^{D \times D}$ denotes the weights for the $l$-th layer.[2] With slight abuse of notation, we denote the vectorization of $\boldsymbol{\Theta}_{L,D}$ as $\mathrm{vec}(\boldsymbol{\Theta}_{L,D})^\top = [\mathrm{vec}(\boldsymbol{W}_1)^\top \;\; \cdots \;\; \mathrm{vec}(\boldsymbol{W}_L)^\top]$.[3] Our goal is to re-use the parameters $\boldsymbol{\Theta} = \boldsymbol{\Theta}_{L_1,D_1}$ from a pretrained smaller model to initialize a large model $\boldsymbol{\Theta}^{(new)} = \boldsymbol{\Theta}_{L_2,D_2}$ through a *model growth operator* $M : \mathbb{R}^{L_1 D_1 \times D_1} \to \mathbb{R}^{L_2 D_2 \times D_2}$ that maps the weights of the smaller network to the weights of the larger one, i.e., $\boldsymbol{\Theta}^{(new)} = M(\boldsymbol{\Theta})$ where $L_1 < L_2$ and $D_1 < D_2$. After model growth, we adopt $\boldsymbol{\Theta}^{(new)}$ as the initialization of the large model and train it using standard recipes.

### 3.1 EXISTING GROWTH OPERATORS

Existing works have separately established model growth operators for depth ($L_1 < L_2, D_1 = D_2$) and width ($L_1 = L_2, D_1 < D_2$). We summarize these methods below.

**Depth expansion.** StackBERT (Gong et al., 2019) proposes to duplicate the smaller model to double the depth, based on the observation that upper layers share similar functionality with the lower layers. In contrast, interpolation-based depth expansion methods (Chang et al., 2017; Dong et al., 2020) interleave every layer to form a deeper model, which can be roughly interpreted as simulating a finer-grained solution to the original dynamical system from a neural ODE perspective (Chen et al., 2018). Letting $L_2 = kL_1$, the two methods' growth operators can be formulated as:

$$\text{(StackBERT) } \boldsymbol{W}_l^{(new)} = \boldsymbol{W}_{l \bmod L_1}, \quad \text{(Interpolation) } \boldsymbol{W}_i^{(new)} = \boldsymbol{W}_{\lfloor l/k \rfloor}, \quad \forall l \in [L_2]. \quad (1)$$

**Width expansion.** Net2Net (Chen et al., 2015) expands the width of neural networks by randomly copying neurons while preserving output values via normalization. This can be seen as growing a matrix associated with a particular layer by duplicating the columns and rows of its weight matrix. Suppose a layer has weight matrix $\boldsymbol{W}_l \in \mathbb{R}^{D_1 \times D_1}$.[4] To expand it to a matrix $\boldsymbol{W}_l^{(new)} \in \mathbb{R}^{D_2 \times D_2}$

---

[2]For notational brevity we assume that each hidden layer has same number of dimensions $D$, but LiGO can be straightforwardly generalized to layers with different dimensions (e.g., FFN layers of transformers).

[3]We therefore have $\mathrm{vec}(\boldsymbol{\Theta}_{L,D})^\top \in \mathbb{R}^{LD^2}$. Our approach is also agnostic with regard to vectorization order.

[4]We define a single layer as $f_l(\boldsymbol{x}) = \boldsymbol{W}_l \boldsymbol{x} + \boldsymbol{b}_l$, where the row number of $\boldsymbol{W}_l$ corresponds to the output dimension, and the column number of $\boldsymbol{W}_l$ corresponds to the input dimension.

$(D_2 > D_1)$, Net2Net copies $\boldsymbol{W}_l$ to its upper-left corner of $\boldsymbol{W}_l^{(new)}$, fills the new columns via a random selection matrix $\boldsymbol{S}_l$, and finally duplicates and normalizes rows according to the selection matrix from the previous layer. Formally, the growth operator of Net2Net can be written as:

$$\text{(Net2Net)} \quad \boldsymbol{W}_l^{(new)} = \begin{bmatrix} \boldsymbol{I} \\ \boldsymbol{S}_{l-1}^\top \end{bmatrix} \boldsymbol{D}_l^{-1} \boldsymbol{W}_l \begin{bmatrix} \boldsymbol{I} & \boldsymbol{S}_l \end{bmatrix}, \quad \boldsymbol{D}_l = \text{diag}(\boldsymbol{S}_{l-1}\boldsymbol{1}) + \boldsymbol{I}, \quad \forall l \in [L_2] \quad (2)$$

where $\boldsymbol{S}_l \in \{0,1\}^{D_1 \times (D_2 - D_1)}$ is a random selection matrix. The diagonal of $\boldsymbol{D}_l$ is a $D_1$-dimensional histogram, whose $i$-th entry indicates number of times $i$-th column of $\boldsymbol{W}_l$ was copied.

## 3.2 LEARNING TO GROW WITH A STRUCTURED LINEAR GROWTH OPERATOR

While existing operators have been empirically successful in accelerating transformer-based models such as BERT (Gong et al., 2019; Chen et al., 2021), we observe that generally do not have a learning component and perform the depth- and width-expansions separately. In this section we introduce a general framework for learning to grow with a linear growth operator (LiGO), which generalizes existing operators by combining the width- and depth-growth operators in a data-driven way.

We can formulate the problem of initializing the weights of the larger model $\Theta^{(new)}$ from the smaller model $\Theta$ through the following optimization problem,

$$\arg\min_M \mathbb{E}_{\boldsymbol{x} \sim \mathcal{D}} \; \mathcal{L}(\boldsymbol{x}; \Theta^{(new)}), \quad \text{subject to } \Theta^{(new)} = M(\Theta), \quad (3)$$

where $\mathcal{D}$ is the data distribution and $\mathcal{L}$ is the loss function. It is of course intractable to optimize over the entire operator space, and thus we further simplify the function $M$ to be a linear transformation, which results in the following formulation,

$$\text{vec}(\Theta^{(new)}) = \text{vec}(M(\Theta)) = \boldsymbol{M}\,\text{vec}(\Theta), \quad \boldsymbol{M} \in \mathbb{R}^{L_2 D_2^2 \times L_1 D_1^2}. \quad (4)$$

This simplified objective is still completely infeasible to apply to contemporary neural networks where $L_1 D_1$ can easily be in the hundreds of millions. We therefore propose an efficient parameterization of $\boldsymbol{M}$ for tractable learning.

### 3.2.1 DECOMPOSITION ALONG DEPTH AND WIDTH

Our first step is to decompose the LiGO operator as $\boldsymbol{M} = \boldsymbol{L}_{depth}\boldsymbol{R}_{width}$, where $\boldsymbol{L}_{depth}$ and $\boldsymbol{R}_{width}$ expand the depth and width of model separately. Concretely, we decompose $\boldsymbol{M}$ as

$$\boldsymbol{M} = \underbrace{\begin{bmatrix} \text{diag}(\boldsymbol{\ell}_{1,1}) & \cdots & \text{diag}(\boldsymbol{\ell}_{1,L_1}) \\ \vdots & \ddots & \vdots \\ \text{diag}(\boldsymbol{\ell}_{L_2,1}) & \cdots & \text{diag}(\boldsymbol{\ell}_{L_2,L_1}) \end{bmatrix}}_{\boldsymbol{L}_{depth}} \underbrace{\begin{bmatrix} \boldsymbol{R}_1 & & \\ & \ddots & \\ & & \boldsymbol{R}_{L_1} \end{bmatrix}}_{\boldsymbol{R}_{width}}. \quad (5)$$

where $\boldsymbol{R}_l \in \mathbb{R}^{D_2^2 \times D_1^2}$ and $\boldsymbol{\ell}_{i,j} \in \mathbb{R}^{D_2^2}$. In the above, $\boldsymbol{L}_{depth}$ is an array of diagonal matrices and $\boldsymbol{R}_{width}$ is a block-diagonal matrix, i.e., both matrices are highly structured and sparse. When applying $\boldsymbol{R}_{width}$ to weights $\text{vec}(\Theta)$, the parameters of each layer will be transformed independently via $\text{vec}(\boldsymbol{W}_l^{(new)}) = \boldsymbol{R}_l\,\text{vec}(\boldsymbol{W}_l)$ and lifted to a higher dimension. The $l$-th row block of $\boldsymbol{L}_{depth}$ corresponds to the growth operator of $l$-th layer, which amounts to linearly combining all layers of the smaller model via $\text{vec}(\boldsymbol{W}_l^{(new)})_k = \sum_{l'=1}^{L_1} (\boldsymbol{\ell}_{l,l'})_k\,\text{vec}(\boldsymbol{W}_l)_k$. By this factorization, we can effectively reduce the complexity of the LiGO operator from $\mathcal{O}(D_1^2 L_1 D_2^2 L_2)$ to $\mathcal{O}(D_1^2 D_2^2 L_1)$ and encode architectural knowledge by grouping parameters by layers. Later in Section 3.4, this representation is also shown to preserve high representation power owing to its connection with Monarch matrices (Dao et al., 2022; 2019).

### 3.2.2 PARAMETER SHARING VIA KRONECKER FACTORIZATION

The above LiGO operator requires $\mathcal{O}(D_1^2 D_2^2 L_1)$ parameters for $\boldsymbol{R}_{width}$ and $\mathcal{O}(L_1 L_2 D_2^2)$ for $\boldsymbol{L}_{depth}$. The width operator $\boldsymbol{R}_{width}$ is thus still prohibitively expensive given that $D_1$ (and $D_2$) can easily be in the hundreds or thousands. In this section, we propose a Kronecker factorization to further reduce the number of learnable parameters for each growth operator.

**Depth.** For depth, we treat an entire layer as a single group and construct a new layer by combining existing layers, effectively tying parameters for all neurons in same layer. Formally, each block in $\boldsymbol{L}_{depth}$ is simplified to be $\text{diag}(\boldsymbol{\ell}_{i,j}) = w_{i,j}\boldsymbol{I}$. Then the entire matrix can be written as a Kronecker factorization, $\boldsymbol{L}_{depth} = \boldsymbol{w} \otimes \boldsymbol{I}$, where $\boldsymbol{w} \in \mathbb{R}^{L_2 \times L_1}$ is a matrix whose entry $w_{i,j}$ indicates blending weights of $j$-th layer of the small model to form $i$-th layer of the large model. This strategy reduces the number of parameters in $\boldsymbol{L}_{depth}$ to $\mathcal{O}(L_1 L_2)$, and is shown on left-hand side of Figure 1.

**Width.** For width, we decompose each diagonal block of width expansion operator $\boldsymbol{R}_{width}$ using the Kronecker factorization $\boldsymbol{R}_l = \boldsymbol{A}_l \otimes \boldsymbol{B}_l$, where $\boldsymbol{A}_l, \boldsymbol{B}_l \in \mathbb{R}^{D_2 \times D_1}$. Since $\text{vec}(\boldsymbol{CAB}) = (\boldsymbol{B}^\top \otimes \boldsymbol{C})\text{vec}(\boldsymbol{A})$ (Schacke, 2004), we then have,

$$\boldsymbol{R}_{width}\text{vec}(\boldsymbol{\Theta}) = \begin{bmatrix} \boldsymbol{A}_1 \otimes \boldsymbol{B}_1 & & \\ & \ddots & \\ & & \boldsymbol{A}_{L_1} \otimes \boldsymbol{B}_{L_1} \end{bmatrix} \text{vec}(\boldsymbol{\Theta}) \qquad (6)$$

$$= \text{vec}\left(\begin{bmatrix} \boldsymbol{B}_1 \boldsymbol{W}_1 \boldsymbol{A}_1^\top & \cdots & \boldsymbol{B}_{L_1} \boldsymbol{W}_{L_1} \boldsymbol{A}_{L_1}^\top \end{bmatrix}^\top\right). \qquad (7)$$

Here we observe that $\boldsymbol{B}_l \boldsymbol{W}_l \boldsymbol{A}_l^\top$ performs in- and out-dimension expansion by $\boldsymbol{A}_l$ and $\boldsymbol{B}_l$, respectively. Each new column/row is a linear combination of columns/rows of small model's weight matrix. This factorization, which can be seen as grouping parameters by *neurons*, reduces the number of parameters to $\mathcal{O}(L_1 D_1 D_2)$. Figure 1 (right) illustrates LiGO's width-expansion operator.

Altogether, we obtain the final parameterization of LiGO operator $\boldsymbol{M}$:

$$\boldsymbol{M} = \underbrace{\left(\begin{bmatrix} w_{1,1} & w_{1,2} & \cdots & w_{1,L_1} \\ \vdots & \vdots & \ddots & \vdots \\ w_{L_2,1} & w_{L_2,2} & \cdots & w_{L_2,L_1} \end{bmatrix} \otimes \boldsymbol{I}\right)}_{\text{Depth expansion}} \underbrace{\left(\begin{bmatrix} \boldsymbol{A}_1 \otimes \boldsymbol{B}_1 & & \\ & \ddots & \\ & & \boldsymbol{A}_{L_1} \otimes \boldsymbol{B}_{L_1} \end{bmatrix}\right)}_{\text{Width expansion}} \qquad (8)$$

We can exploit the factorization to implement the LiGO operator (Eq. 8) efficiently.

**Training.** LiGO expands a model in three steps: (1) for each layer, inserting new rows by linearly combining existing rows through $\boldsymbol{B}_l$, (2) for each layer, inserting new columns by linearly combining existing columns through $\boldsymbol{A}_l$, and then finally (3) reconstructing each layer by linearly combining the weight matrices with $\boldsymbol{w}$ along the depth. We then run a few steps (e.g., 100 iterations) of SGD to optimize $\boldsymbol{M}$, which has negligible compute cost relative to regular training. After obtaining $\boldsymbol{M}$, we initialize large model with $\boldsymbol{M}\,\text{vec}(\boldsymbol{\Theta})$, and train parameters $\boldsymbol{\Theta}^{(new)}$ through SGD as usual. Algorithm 1 summarizes a forward pass of LiGO with transformer. Finally, as shown in Appendix A we note that StackBERT (Eq. 1), Interpolation (Eq. 1), and Net2Net (Eq. 2) are all special cases of LiGO (Eq. 8) with a particular setting of $\boldsymbol{L}_{depth}$ and $\boldsymbol{R}_{width}$.

### 3.3 LiGO for Transformers

While LiGO can be applied to any multi-layer neural network architecture, in this paper we focus on using LiGO to grow transformers which have been shown to be particularly amenable to scaling. Below we briefly describe how LiGO is applied to the main transformer embedding/attention layers and defer further details (e.g., growing bias vectors, layer norm parameters) to Appendix B.1.

**Embedding layer.** The embedding layer can be regarded as a linear layer whose inputs are one-hot vectors. We learn a matrix $\boldsymbol{B}^{(emb)}$ to extend its output dimension. This embedding layer is also used as the final output layer for our transformer language modeling experiments.

**Attention and feedforward Layers.** An attention layer consists of multi-head attention weights $(\boldsymbol{W}^Q, \boldsymbol{W}^K, \boldsymbol{W}^V)$ and a linear projection $(\boldsymbol{W}^O)$. Let $\boldsymbol{A}_l^k$ and $\boldsymbol{B}_l^k$ where $k \in \{Q, K, V, O\}$ be the $l$-th layer's in- and out-dimension expansion matrices (Eq. 6) for the query, key, value, and projection matrices. To make sure new input and output channels are aligned across modules, we tie the LiGO operator as follows: for all $l \in [L_1]$, (1) $\boldsymbol{A}_l^k = (\boldsymbol{B}^{(emb)})^\top$ for $\forall k \in \{Q, K, V\}$, (2) $\boldsymbol{A}_l^O = (\boldsymbol{B}_l^V)^\top$, (3) $\boldsymbol{B}_l^O = \boldsymbol{B}^{(emb)}$. The last constraint is added to take into account the residual connections (Chen et al., 2021). We similarly tie parameters for the feed-forward networks, $\boldsymbol{A}_l^{(fc1)} = (\boldsymbol{B}^{(emb)})^\top$, $\boldsymbol{A}_l^{(fc2)} = (\boldsymbol{B}^{(fc1)})_l^\top$ and $\boldsymbol{B}_l^{(fc2)} = \boldsymbol{B}^{(emb)}$. Since transformers make heavy use of residual layers

with skip connections, we found that simply using the same $\boldsymbol{B}^{(emb)}$ to parameterize $\boldsymbol{A}_l^k$ and $\boldsymbol{B}_l^k$ for many layers/modules worked well in practice. This reduces the number of learnable parameters even further and enables fast learning of $\boldsymbol{M}$ on a small amount of data (100 gradient steps).

### 3.4 CONNECTION TO MONARCH MATRICES

As shown in Section 3.2.1, our depth-width decomposition factorizes $\boldsymbol{M}$ into a multiplication of two structured sparse matrices. We examine the expressiveness of this factorized representation by relating it to Monarch matrices (Dao et al., 2022), defined below.

**Definition 1.** *Let the space of Monarch matrices be $\mathcal{M} \subseteq \mathbb{R}^{mn_1 \times mn_2}$. Then matrix $\boldsymbol{M} \in \mathcal{M}$ if $\boldsymbol{M} = \boldsymbol{P}_1 \boldsymbol{L} \boldsymbol{P}_2^\top \boldsymbol{R} = \boldsymbol{P}_1 \operatorname{diag}(\boldsymbol{L}_1, \cdots, \boldsymbol{L}_{n_1}) \boldsymbol{P}_2^\top \operatorname{diag}(\boldsymbol{R}_1, \cdots, \boldsymbol{R}_{n_2})$ where $\boldsymbol{L}_i \in \mathbb{R}^{b_1 \times b_2}$, $\boldsymbol{R}_i \in \mathbb{R}^{b_3 \times b_4}$ are dense rectangular matrices, and $n_1 b_2 = n_2 b_3$. $\boldsymbol{P}_1$ is the permutation $\pi(i) = (i - b_1 \lfloor \frac{i}{b_1} \rfloor - 1) n_1 + \lfloor \frac{i}{b_1} \rfloor + 1$ and $\boldsymbol{P}_2$ is the permutation $\pi(j) = (j - b_2 \lfloor \frac{j}{b_2} \rfloor - 1) n_1 + \lfloor \frac{j}{b_2} \rfloor + 1$.*

It is clear that the block-diagonal matrix $\boldsymbol{R}$ has the identical form to our width growing operator $\boldsymbol{R}_{width}$. By applying the permutation matrices $\boldsymbol{P}_1$ and $\boldsymbol{P}_2$ to $\boldsymbol{L}$, $\boldsymbol{L}$ is transformed into exactly the same form with our depth-growth operator $\boldsymbol{L}_{depth}$ in Eq. 5. This implies that our depth-width decomposition coincides with Monarch sparsification of dense matrices, which generalize butterfly matrices (Dao et al., 2019) and enjoy rich expressivity properties (Dao et al., 2020; 2022).

## 4 EXPERIMENTS

We conduct experiments to answer three key research questions. Q1: To what extent can LiGO improve the training efficiency (FLOPs and wall time) of transformers compared to training from scratch and other growth operators? Q2: Can LiGO be universally effective across transformers from different domains (e.g., language and vision) and sizes? Q3: Can models trained using LiGO achieve similar performance compared to the baselines when transferred to downstream tasks?

### 4.1 EXPERIMENTAL SETUP

**Datasets.** We follow Tan & Bansal (2020) and use the English Wikipedia corpus[5] for training BERT (Devlin et al., 2019) and RoBERTa (Liu et al., 2019). We use the public C4 (Raffel et al., 2020) dataset for training GPT2 (Radford et al., 2019). We use ImageNet (Deng et al., 2009) for training vision transformers. We use GLUE (Wang et al., 2018), SQuADv1.1 (Rajpurkar et al., 2016), and SQuADv2.0 (Rajpurkar et al., 2018) for evaluating pretrained BERT models. We test downstream performance of vision transformers (DeiT (Touvron et al., 2021a)) by performing transfer learning on 5 downstream image classification tasks, including CIFAR10 (Krizhevsky et al., 2009), CIFAR100 (Krizhevsky et al., 2009), Flowers102 (Nilsback & Zisserman, 2008), Stanford-Cars (Krause et al., 2013), and ChestXRay8 (Wang et al., 2017).

**Models.** We experiment with growing the following language andd vision transformers: (1) BERT-Small→BERT-Base, BERT-Base→BERT-Large, BERT-Small→BERT-Large; (2) RoBERTa-Small→RoBERTa-Base for RoBERTa; (3) GPT2-Base→GPT2-Medium, (4) DeiT-S→DeiT-B, and (5) CaiT-XS→CaiT-S. BERT-Small has 6 layers with 512 hidden dimensions, while other named models are their usual sizes. See Appendix B.2 for full details.

**Baselines.** We compare our approach with the following baselines: (1) training from scratch baseline where we train the larger transformer without using any smaller pretrained models; (2) progressive training methods designed for growing depth in transformers (StackBERT (Gong et al., 2019) and MSLT (Yang et al., 2020)); (3) bert2BERT (Chen et al., 2021) that extends Net2Net (Chen et al., 2015) for width expansion and stacking for depth expansion; (4) KI (Qin et al., 2021) which uses distillation for transferring knowledge from the smaller model to the larger model.

**Implementation details.** We always use 100 gradient steps to learn the LiGO for all models, which is negligible in terms of FLOPs/wall time compared to full training after initialization. We train both BERT and RoBERTa models for 400K steps with a warmup of 10K steps. We remove the next-sentence prediction task (Liu et al., 2019) and use a fixed sequence length of 128 for pretraining

---

[5]While the original BERT (Devlin et al., 2019) paper also uses the Toronto Book Corpus (Zhu et al., 2015), we do not include it here since it is no longer publicly available.

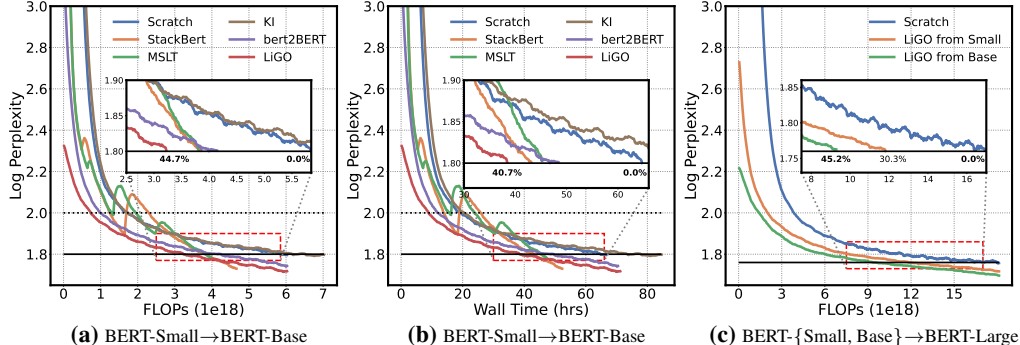

**Figure 2:** Results on BERT. (a-b) shows validation log perplexity vs. FLOPs and wall time respectively for training BERT-Base by reusing BERT-Small. (c) shows log perplexity vs. FLOPs in growing BERT-Small and BERT-Base to BERT-Large. The solid line indicates the final perplexity of the larger model trained from scratch, while the dotted line represents performance of the smaller model trained from scratch. LiGO offers about 45% savings in FLOPs and 40% savings in wall time over BERT-Base training from scratch. Our approach is also flexible in reusing either BERT-Small or BERT-Base for accelerating BERT-Large training.

**Table 1:** Downstream transfer learning performance on GLUE and SQuAD. All of the results are based on BERT-Base models trained using the different baselines. LiGO achieves similar or even better performance than the original training from scratch baseline on several downstream tasks, despite improving training efficiency.

| Method | Savings (FLOPs) | Savings (Walltime) | SST-2 (Acc.) | MNLI (Acc.) | MRPC (Acc.) | CoLA (Acc.) | QNLI (Acc.) | QQP (Acc.) | STS-B (Acc.) | SQuADv1.1 (F1/EM) | SQuADv2.0 (F1/EM) | Avg. GLUE | Avg. SQuAD |
|---|---|---|---|---|---|---|---|---|---|---|---|---|---|
| Scratch | – | – | 88.19 | 78.43 | 85.78 | 62.09 | 87.06 | 87.18 | 86.99 | 86.55 / 77.31 | 71.31 / 67.07 | 82.25 | 78.79 / 72.19 |
| StackBERT | 34.1% | 33.3% | 88.99 | 79.72 | 85.29 | 59.09 | 87.28 | 89.17 | 86.97 | 86.50 / 77.42 | 71.32 / 67.41 | 82.36 | 78.91 / 72.41 |
| MSLT | 34.9% | 30.0% | 88.53 | 78.10 | 82.60 | 64.65 | 83.58 | 88.54 | 85.89 | 86.07 / 76.73 | 70.68 / 67.17 | 81.72 | 78.47 / 71.95 |
| KI | -5.7% | -13.9% | 88.65 | 78.83 | 83.50 | 64.86 | 86.25 | 88.96 | 87.09 | 84.93 / 76.29 | 71.09 / 67.41 | 82.59 | 78.01 / 71.85 |
| bert2BERT | 29.0% | 25.1% | 88.30 | 80.05 | 85.54 | 61.73 | 88.16 | 86.18 | 87.00 | 86.24 / 77.09 | 71.52 / 66.85 | 82.42 | 78.88 / 71.97 |
| LiGO | 44.7% | 40.7% | 88.42 | 79.29 | 84.31 | 62.09 | 88.07 | 88.81 | 87.00 | 86.28 / 77.45 | 71.24 / 67.17 | 82.57 | 78.76 / 72.31 |

both models. For BERT, we use a batch size of 256 and a learning rate of $2e^{-4}$, while we use a batch size of 1024 and a learning rate of $8e^{-4}$ for training RoBERTa models.

Following Shen et al. (2022), we train GPT2 models with a batch size of 384 and sequence length of 1024. For vision transformers, we build our models based on DeiT (Touvron et al., 2021a) and CaiT (Touvron et al., 2021b), and apply their default hyper-parameters for training on ImageNet dataset. We train all our vision transformers for 300 epochs with a batch size of 1024. For transfer learning with BERT/RoBERTa, we follow Tan & Bansal (2020) and train for 3 epochs with a learning rate of $1e^{-4}$ and a batch-size of 32 for all tasks in GLUE. On SQuAD v1.1 and SQuAD 2.0, we fine-tune for 2 epochs with a learning rate of $5e^{-5}$ and a batch size of 12. We run both GLUE and SQuAD evaluations three times with different random seeds and report the mean numbers. For transfer learning experiments on DeiT, we finetune the pretrained models with 1000 epochs, batch size 768, learning rate 0.01, and use the same data augmentation in training on ImageNet. We use the same pretraining data and experimental settings for all the baselines (including our approach) for a fair comparison. Note that we include the additional compute required for training LiGO in all our tables and figures. However, since our LiGO is only trained for 100 steps, the influence on visualization and quantitative saving percentages is negligible.

## 4.2 RESULTS AND ANALYSIS

**BERT.** Figure 2 shows the comparison between the different baselines for training BERT models. As seen from Figure 2(a), LiGO saves 44.7% computational cost (FLOPs) of training BERT-Base (12 layers, 768 dimensions) from scratch by reusing BERT-Small (6 layers, 512 dimensions). LiGO offers 40.7% savings in wall time compared to training from scratch (Figure 2(b)). Among the compared methods, StackBERT is the most competitive in terms of both FLOPs and wall time, although LiGO obtains +10.6% and +7.2% improvements in FLOPs and wall time on top of StackBERT. Similarly, LiGO significantly outperforms the recent bert2BERT method which saves about 30% computational costs. We observe that KI does not provide any real savings in training as it requires additional computation for knowledge distillation. Figure 2(c) shows that our LiGO approach is flexible in growing either BERT-Small or BERT-Base for accelerating BERT-Large training. As expected, reusing BERT-Base instead of BERT-Small leads more savings in FLOPs (45.2% vs 30.3%) as BERT-Base contains more implicit knowledge in its parameters. Table 1 shows the per-task per-

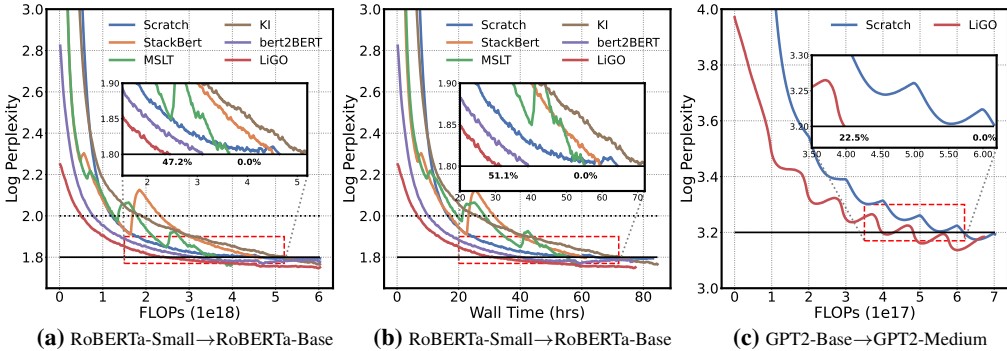

**Figure 3:** Results on RoBERTa and GPT2. LiGO reduces FLOPs by $47.2\%$ and $22.5\%$ for RoBERTa-Base and GPT2-Medium, , demonstrating its effectiveness across different training strategies and architectures.

formance of different BERT-Base models on both GLUE and SQuAD benchmarks, where we find that BERT trained with LiGO achieves very similar performance compared to the baselines on both benchmarks. Finally, in Table 5 of the Appendix C.3, we show that growing BERT-Small to BERT-Base with 100 steps of LiGO and then finetuning on GLUE tasks *without* additional pretraining outperforms just directly finetuning BERT-Small.

**RoBERTa and GPT2.** Figure 3(a-b) shows the results on RoBERTa, whose training recipe uses larger batch size and learning rate than BERT. LiGO similarly accelerates RoBERTa training, which indicates that our method is robust to optimization hyperparameters. On GPT2, LiGO saves $22.5\%$ computation cost of training GPT2-Medium (345M parameters) by reusing GPT2-Base (117M parameters) (Figure 3(c)). These consistent improvements show that LiGO is effective for accelerating transformer training across different model architectures and sizes.

**Vision Transformers.** Figure 4 shows that by growing from DeiT-S, LiGO can save $55.4\%$ FLOPs and $52\%$ GPU wall time to reach the same performance of $81\%$ on ImageNet. Interestingly, the model initialized by our data-driven growth operator (w/ only 100 gradient steps of tuning) can already achieve $72\%$ accuracy at the beginning of training and leads to the final accuracy of $81.7\%$ at the end of the training. Compared to the next best method, bert2BERT, LiGO

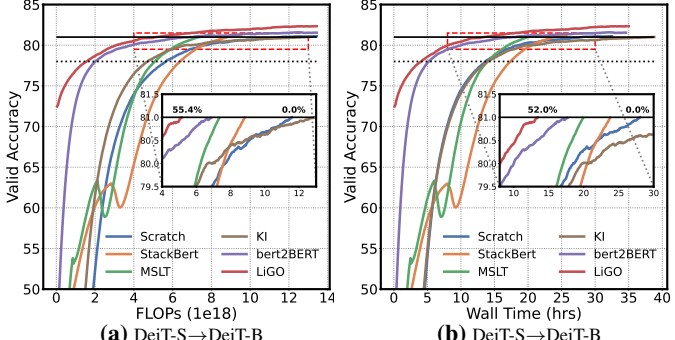

**Figure 4:** Results on DeiT. (a) Accuracy vs. flops and (b) accuracy vs. wall time for training DeiT-B. LiGO saves flops and wall time by more than 50% over training from scratch on ImageNet.

obtains more than $15\%$ savings, which once again demonstrates the effectiveness of our approach in growing vision transformers as well. Table 2 shows that finetuning results on downstream tasks perform on-par with the model trained from scratch, showing that LiGO does not harm the model's generalization capabilities when transferred to downstream datasets. We also find similar savings in CaiT-XS→CaiT-S where LiGO saves FLOPs by $52.6\%$ and wall time by $46.1\%$ over training CaiT-S from scratch on ImageNet (see Appendix C.2 for more details).

**Combining with other training strategies.** We also find that LiGO can be effectively combined with orthogonal strategies such as layer dropping (Zhang & He, 2020), token dropping (Hou et al., 2022), and staged training (Chen et al., 2021). More details are included in Appendix B.3. Figure 5 shows that LiGO can

**Table 2:** Transfer learning performance of DeiT-B. DeiT-B model trained using LiGO performs similarly to the original train-from-scratch baseline on all downstream tasks.

| Method | FLOPs | Walltime | ImageNet | CIFAR10 | CIFAR100 | Flowers | Cars | ChestXRay8 |
|---|---|---|---|---|---|---|---|---|
| Scratch | – | – | 81.10 | 99.09 | 90.76 | 97.79 | 92.06 | 55.81 |
| StackBERT | 23.8% | 15.1% | 81.21 | 99.11 | 90.80 | 97.56 | 92.09 | 55.77 |
| MSLT | 36.7% | 28.9% | 81.27 | 99.07 | 90.21 | 97.71 | 92.11 | 55.79 |
| KI | −11.2% | −36.8% | 81.01 | 98.94 | 90.32 | 97.81 | 92.08 | 55.80 |
| bert2BERT | 40.8% | 37.0% | 81.59 | 99.14 | 90.69 | 97.67 | 92.15 | 55.82 |
| LiGO | 55.4% | 52.0% | 81.71 | 99.12 | 90.74 | 97.77 | 92.09 | 55.82 |

be combined with other training techniques to improve the computational savings by $4.7\%$, $7.4\%$,

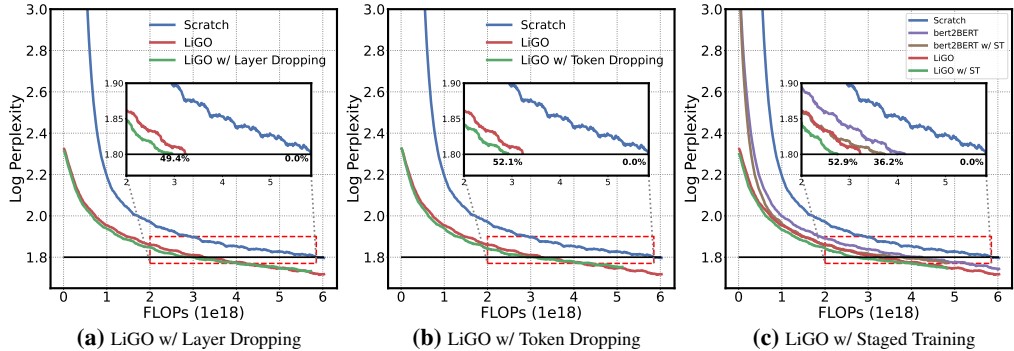

**Figure 5:** LiGO with other efficient training strategies. Our approach can be combined with (a) layer dropping, (b) token dropping, and (c) staged training (ST), for further accelerate BERT training.

and $8.2\%$ with layer dropping, token dropping and staged training, respectively. Following (Chen et al., 2021), we also apply staged training strategy to bert2BERT and observe that LiGO still outperforms bert2BERT with staged training by $16.7\%$ (see Figure 5(c)).

### 4.3 ABLATION STUDIES

**Depth-only expansion.** We examine the effectiveness of our proposed depth expansion operator ($\boldsymbol{L}_{depth}$) by only growing the depth of BERT from 6 layers to 12 layers, i.e, (BERT(6, 768)→BERT(12, 768). We compare with stacking (Stack-BERT, Gong et al., 2019), Interpolation (InterBERT, Chang et al., 2017; Dong et al., 2020) (see Eq. 1), and MSLT (Yang et al., 2020). For LiGO, we only apply its $\boldsymbol{L}_{depth}$ component to the pre-trained model weights. Results in Figure 6(a) show that a data-driven approach works well even when just growing across the depth dimension.

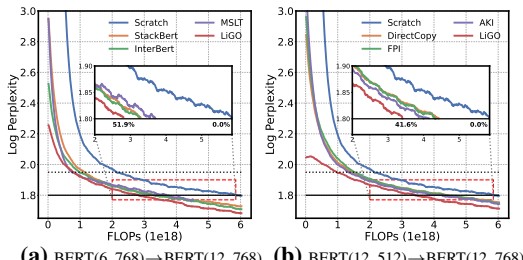

**(a)** BERT(6, 768)→BERT(12, 768) **(b)** BERT(12, 512)→BERT(12, 768)

**Figure 6:** Results on Depth-only and Width-only growth. LiGO saves $51.7\%$ FLOPS when expanding depth-only, and $41.6\%$ FLOPS when expanding width-only.

**Width-only expansion.** We also verify the effectiveness of $\boldsymbol{R}_{width}$ by only extending BERT width from 512 to 768, i.e., BERT(12, 512)→BERT(12, 768). We compare LiGO based initialization with direct copy (Wei et al., 2016), function preserving initialization (FPI, Chen et al., 2015), and advanced knowledge initialization (AKI, Chen et al., 2021). LiGO's width expansion component outperforms all other methods, as shown in Figure 6(b).

**Table 3:** Effect of number of gradient steps. "+FLOPs" stands for additional flops (in $10^{15}$).

| # of Steps | +FLOPs | Savings |
|---|---|---|
| 100 | 3.61 | 44.7% |
| 500 | 18.06 | 44.5% |
| 1000 | 36.13 | 44.2% |
| 10000 | 361.30 | 38.9% |

**Number of growing steps.** Our main experiments just use 100 gradient steps to grow. We tune our LiGO on the pretraining set for 100, 500, 1000, and 10000 steps and compute the additional FLOPs for BERT-Small→BERT-Base training. Table 3 shows that training LiGO within 1000 steps results in the identical model convergence (reaching 1.8 PPL at 215K steps). This suggests tuning model weights under the linear constraints of LiGO can achieve faster convergence. Training LiGO for more than 10000 steps can provide a model with slightly faster convergence (214K steps), but results in less saving overall.

## 5 CONCLUSION

This paper describes an approach for accelerating transformer training by learning to grow pretrained transformers, where the larger transformer's parameters are initialized as a linear mapping from the smaller pretrained model's parameters, The linear map is factorized to be a composition of sparse width- and depth-expansion operators with a Kronecker factorization that groups parameters into layers and neurons. We demonstrate the effectiveness of our proposed approach on both language and vision transformers of different sizes, outperforming several competing methods. While our compute resources prevented us from applying LiGO to even larger transformers, it would be interesting to see if this can be applied on top of even larger models.

ACKNOWLEDGMENTS

PW sincerely thanks Zhen Wang for the insightful discussion and for providing reference repositories for language model pre-training. PW also appreciates Hao Tan's assistance for reproducing fine-tuning results on GLUE datasets. YK and LTH were partially supported an MIT-IBM Watson AI grant and an Amazon award. We also acknowledge support from the IBM Research AI Hardware Center, and the Center for Computational Innovation at Rensselaer Polytechnic Institute for the computational resources on the AiMOS Supercomputer. The research of ZW is in part supported by the US Army Research Office Young Investigator Award (W911NF2010240).

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

# A    UNIVERSALITY OF LiGO OPERATOR

**Proposition 1.** *StackBERT (Eq. 1), Interpolation (Eq. 1), and Net2Net (Eq. 2) are all the special cases of the LiGO operator (Eq. 8).*

*Proof.* We prove Proposition 1 by constructing parameters in $\boldsymbol{L}_{depth}$ and $\boldsymbol{R}_{width}$.

**Stacking.** Stacking-based methods (Gong et al., 2019; Yang et al., 2020) duplicate the entire lower blocks on top of the small model to the form new layers (Eq. 1). Formally, we show this operation can be done by the following operator:

$$
\boldsymbol{M} = \underbrace{\begin{bmatrix} \boldsymbol{I} & & & \\ & \boldsymbol{I} & & \\ & & \ddots & \\ \boldsymbol{I} & & & \\ & \boldsymbol{I} & & \\ & & & \ddots \end{bmatrix}}_{\boldsymbol{L}_{depth}} \underbrace{\begin{bmatrix} \boldsymbol{I} & & \\ & \ddots & \\ & & \boldsymbol{I} \end{bmatrix}}_{\boldsymbol{R}_{width}} \tag{9}
$$

**Interpolation.** Interpolation based methods (Chang et al., 2017; Dong et al., 2020) interleave each layer for twice. We can construct the following matrix to achieve layer interpolation (Eq. 1).

$$
\boldsymbol{M} = \underbrace{\begin{bmatrix} \boldsymbol{I} & & & \\ \boldsymbol{I} & & & \\ & \boldsymbol{I} & & \\ & \boldsymbol{I} & & \\ & & \ddots & \\ & & & \ddots \end{bmatrix}}_{\boldsymbol{L}_{depth}} \underbrace{\begin{bmatrix} \boldsymbol{I} & & \\ & \ddots & \\ & & \boldsymbol{I} \end{bmatrix}}_{\boldsymbol{R}_{width}} \tag{10}
$$

We remark that any rearrangement of layers to construct new layers (mathematically a permutation of existing layers with replacement) can be constructed in a similar way.

**Net2Net.** Since we show in Eq. 6, the Kronecker factorization on $\boldsymbol{R}_l$ amounts to decomposing the general growth operator into in-dimension and out-dimension expansion. We can construct Net2Net (Chen et al., 2015) based growth by simply letting:

$$
\boldsymbol{L}_{depth} = \boldsymbol{I} \in \mathbb{R}^{L_1 D_2 \times L_2 D_2}, \quad \boldsymbol{R}_{width} = \begin{bmatrix} \boldsymbol{A}_1 \otimes \boldsymbol{B}_1 & & \\ & \ddots & \\ & & \boldsymbol{A}_{L_1} \otimes \boldsymbol{B}_{L_1} \end{bmatrix} \tag{11}
$$

$$
\boldsymbol{A}_l = \begin{bmatrix} \boldsymbol{I} \\ \widetilde{\boldsymbol{S}}_{l-1} \end{bmatrix}, \quad \boldsymbol{B}_l = \begin{bmatrix} \boldsymbol{I} \\ \boldsymbol{S}_{l,} \end{bmatrix} \tag{12}
$$

where $\boldsymbol{S}_l \in \{0, 1\}^{(D_2-D1) \times D_1}$ is a selection matrix to enlarge the out dimension, and $\widetilde{\boldsymbol{S}}_{l-1} = \boldsymbol{S}_{l-1} \operatorname{diag}(\boldsymbol{1}^\top \boldsymbol{S}_{l-1})^{-1}$ copies the selection from $\boldsymbol{S}_{l-1}$ with normalization to guarantee functionality preserving in expansion. □

# B    IMPLEMENTATION DETAILS

## B.1    GROWING TRANSFORMERS WITH LiGO

The transformer architecture consists of an embedding layer, multi-block attention layer, and an output layer. The core ingredient attention block consists of a Multi-Head Attention (MHA) module followed by a FeedForward Network (FFN), with a skip connection across the both blocks. Applying LiGO requires the following considerations:

**Embedding layer.** For both language and vision transformers, the embedding layer can be regarded as a linear layer, whose inputs are one-hot embeddings in language models. We draw a learnable matrix $\boldsymbol{B}^{(emb)}$ to extend its output dimension.

**Multi-head attention blocks.** An attention layer in transformer consists of multi-head attention weights ($\boldsymbol{W}^Q, \boldsymbol{W}^K, \boldsymbol{W}^V$) and a linear projection ($\boldsymbol{W}^O$). Let $\boldsymbol{A}_l^k$ and $\boldsymbol{B}_l^k$ with $k \in \{Q, K, V, O\}$ be the in- and out-dimension expansion matrices (Eq. 6) for query, key, value, and projection in the $l$-th layer, respectively. Applying $\boldsymbol{B}_l^k$ to $\boldsymbol{W}^k$ ($k \in Q, K, V$) constructs new heads by a weighted summation of rows of all existing heads. To make sure the new input and output channels are aligned across modules, we tie our LiGO operator with the following scheme: (1) $\boldsymbol{A}_l^k = (\boldsymbol{B}^{(emb)})^\top$ for $\forall k \in \{Q, K, V\}$, (2) $\boldsymbol{A}_l^O = (\boldsymbol{B}_l^{(V)})^\top$, (3) $\boldsymbol{B}_l^O = \boldsymbol{B}^{(emb)}$ for $\forall l \in [L_1]$. Both the bias and layer normalization inherit the associated linear transformations' out-dimension expansion matrices to grow the width. For depth expansion, each module independently combines the same module from other layers (Eq. 8) with learnable coefficients $\boldsymbol{w}$.

**Feed-forward networks.** Each attention block is followed by a two-layer FFN. Let $\boldsymbol{A}_l^k$ and $\boldsymbol{B}_l^k$ with $k \in \{fc1, fc2\}$ be the in- and out-dimension expansion matrices (Eq. 6) for the first and second FFN layer in the $l$-th layer, respectively. We tie the parameters for feed-forward networks: $\boldsymbol{A}_l^{(fc1)} = \boldsymbol{B}^{(emb)\top}$, $\boldsymbol{A}_l^{(fc2)} = \boldsymbol{B}_l^{(fc1)\top}$ and $\boldsymbol{B}_l^{(fc2)} = \boldsymbol{B}^{(emb)}$.

**Output layer.** For output head, we have $\boldsymbol{A}^{(out)} = \boldsymbol{B}^{(emb)\top}$, since the output dimension of attention layers are always aligned with $\boldsymbol{B}^{(emb)}$ by our construction. The output layer does not need out-dimension expansion. Algorithm 1 summarizes LiGO for growing transformers.

## B.2 MODEL CONFIGURATIONS

We summarize the settings of different transformer models used for our experiments in Table 4. For BERT and RoBERTa, we re-use the code base provided by Tan & Bansal (2020). For GTP2, we follow the model configuration of OpenAI and use the pre-training code provided by Shen et al. (2022). For DeiT, we use their official codebase (Touvron et al., 2021a).

**Table 4:** Configuration of different transformers.

|             | BERT-Small | BERT-Base | RoBERTa-Small | RoBERTa-Base | GPT2-Base | GPT2-Medium |            | DeiT-S | DeiT-B |
|-------------|------------|-----------|---------------|--------------|-----------|-------------|------------|--------|--------|
| # layers    | 6          | 12        | 6             | 12           | 12        | 24          | # layers   | 12     | 12     |
| # hidden    | 512        | 768       | 512           | 768          | 768       | 1024        | # hidden   | 384    | 768    |
| # heads     | 8          | 12        | 8             | 12           | 12        | 16          | # heads    | 6      | 12     |
| # vocab     | 30522      | 30522     | 50265         | 50265        | 50257     | 50257       | input res. | 224    | 224    |
| seq. length | 128        | 128       | 128           | 128          | 1024      | 1024        | patch size | 16     | 16     |

## B.3 ORTHOGONAL EFFICIENT TRAINING STRATEGIES

For layer dropping, we follow the same progressive dropping rate schedule with Zhang & He (2020), and set the maximum dropping rate to 0.1 to recover the performance. For token dropping, we randomly set 15% tokens aside in the middle layers. In the first 50k steps of staged training, only a sub-network is activated and trained, and afterwards, we perform full-model training for 350k steps.

## C ADDITIONAL EXPERIMENTS

### C.1 REUSING SMALLER MODELS TRAINED FOR ONLY FEW STEPS

LiGO focuses on utilizing the knowledge of smaller models that have already been pretrained and available. In this section, we investigate how LiGO can leverage smaller existing models that are only trained for few steps to accelerate training of a larger model. We perform an experiment on BERT-Base by reusing a BERT-Small trained for only 50k steps instead full training for 220k steps as used in our experiments. Figure 7 shows that LiGO can still save 35.2% savings in FLOPs and 30.2% savings in wall time over the BERT-Base training from scratch.

---

**Algorithm 1** A forward pass of LiGO with transformer.

---

1: **Input**: A small transformer with hidden $D_1$ and number of layer $L_1$. Denote the embedding layer as $\boldsymbol{W}^{(emb)} \in \mathbb{R}^{D_1 \times E}$, attention layers as $\boldsymbol{W}_l^Q, \boldsymbol{W}^K, \boldsymbol{W}_l^V, \boldsymbol{W}_l^O \in \mathbb{R}^{D_1 \times D_1}$, FFN layers as $\boldsymbol{W}_l^{(fc1)} \in \mathbb{R}^{4D_1 \times D_1}$, $\boldsymbol{W}_l^{(fc2)} \in \mathbb{R}^{D_1 \times 4D_1}$, LayerNorm layers as $\boldsymbol{W}_l^{(ln1)} \in \mathbb{R}^{D_1 \times 4D_1}$, $\boldsymbol{W}_l^{(ln1)}, \boldsymbol{W}_l^{(ln2)} \in \mathbb{R}^{D_1}, \forall l \in [L_1]$, the output head $\boldsymbol{W}^{(out)} \in \mathbb{R}^{C \times D_1}$

2: **Output**: A large transformer with hidden $D_2$ and number of layer $L_2$. Denote the weight matrices as $\boldsymbol{\Omega}$ with the corresponding superscripts as the small model.

3: $\boldsymbol{\Omega}^{(emb)} \leftarrow \boldsymbol{B}^{(emb)} \boldsymbol{W}^{(emb)}$

4: **for** $l = 1, \cdots, L_1$ **do**             ▷ Width Expansion

5:      $\boldsymbol{\Omega}_l^Q \leftarrow \boldsymbol{B}^{(Q)} \boldsymbol{W}_l^Q \boldsymbol{B}^{(emb)\top}$

6:      $\boldsymbol{\Omega}_l^K \leftarrow \boldsymbol{B}^{(K)} \boldsymbol{W}_l^K \boldsymbol{B}^{(emb)\top}$

7:      $\boldsymbol{\Omega}_l^V \leftarrow \boldsymbol{B}^{(V)} \boldsymbol{W}_l^V \boldsymbol{B}^{(emb)\top}$

8:      $\boldsymbol{\Omega}_l^O \leftarrow \boldsymbol{B}^{(emb)} \boldsymbol{W}_l^V \boldsymbol{B}^{(V)\top}$

9:      $\boldsymbol{\Omega}_l^{(ln1)} \leftarrow \boldsymbol{B}^{(emb)} \boldsymbol{W}_l^{(ln1)}$

10:      $\boldsymbol{\Omega}_l^{(fc1)} \leftarrow \boldsymbol{B}^{(fc1)} \boldsymbol{W}_l^V \boldsymbol{B}^{(emb)\top}$

11:      $\boldsymbol{\Omega}_l^{(fc2)} \leftarrow \boldsymbol{B}^{(emb)} \boldsymbol{W}_l^V \boldsymbol{B}^{(fc1)\top}$

12:      $\boldsymbol{\Omega}_l^{(ln2)} \leftarrow \boldsymbol{B}^{(emb)} \boldsymbol{W}_l^{(ln2)}$

13: **end for**

14: **for** $l = 1, \cdots, L_2$ **do**             ▷ Depth Expansion

15:      $\boldsymbol{\Omega}_l^Q \leftarrow \sum_{j=1}^{L_1} w_{l,j}^Q \boldsymbol{\Omega}_j^Q$

16:      $\boldsymbol{\Omega}_l^K \leftarrow \sum_{j=1}^{L_1} w_{l,j}^K \boldsymbol{\Omega}_j^K$

17:      $\boldsymbol{\Omega}_l^V \leftarrow \sum_{j=1}^{L_1} w_{l,j}^V \boldsymbol{\Omega}_j^V$

18:      $\boldsymbol{\Omega}_l^O \leftarrow \sum_{j=1}^{L_1} w_{l,j}^O \boldsymbol{\Omega}_j^O$

19:      $\boldsymbol{\Omega}_l^{(ln1)} \leftarrow \sum_{j=1}^{L_1} w_{l,j}^{(ln1)} \boldsymbol{\Omega}_j^{(ln1)}$

20:      $\boldsymbol{\Omega}_l^{(fc1)} \leftarrow \sum_{j=1}^{L_1} w_{l,j}^{(fc1)} \boldsymbol{\Omega}_j^{(fc1)}$

21:      $\boldsymbol{\Omega}_l^{(fc2)} \leftarrow \sum_{j=1}^{L_1} w_{l,j}^{(fc2)} \boldsymbol{\Omega}_j^{(fc2)}$

22:      $\boldsymbol{\Omega}_l^{(ln2)} \leftarrow \sum_{j=1}^{L_1} w_{l,j}^{(ln2)} \boldsymbol{\Omega}_j^{(ln2)}$

23: **end for**

24: $\boldsymbol{\Omega}^{(out)} \leftarrow \boldsymbol{W}^{(out)} \boldsymbol{B}^{(emb)\top}$

25: Train transformer with parameters $\boldsymbol{\Omega}$.

---

## C.2   Results on CaiT

In addition to DeiT (Touvron et al., 2021a), we perform additional experiments with CaiT (Touvron et al., 2021b) on ImageNet and find that while reusing CaiT-XS, LiGO offers about 52.6% savings in FLOPs and 46.1% savings in wall time over the CaiT-S training from scratch (see Figure 8).

## C.3   Task-specific finetuning with LiGO initialization without further pretraining

We perform additional experiments by directly finetuning BERT-Base initialized by LiGO (from BERT-Small) without any further pretraining. We observe in Table 5 that the LiGO-initialized model can benefit downstream tasks compared to BERT-Small trained from scratch (1st row vs 2nd row).

## C.4   GLUE Performance using AdapterFusion

LiGO is mainly proposed for improving efficiency of the pre-training stage and hence is compatible with various finetuning schemes like full model finetuning, adapters (Houlsby et al., 2019; Pfeiffer et al., 2020) or prompt tuning (Lester et al., 2021; Jia et al., 2022) for adaptation to downstream tasks. We test BERT-Base models trained using different baselines by using adapterfusion (Pfeiffer et al., 2020) instead of full finetuning on GLUE benchmark. Table 6 shows that LiGO also achieves

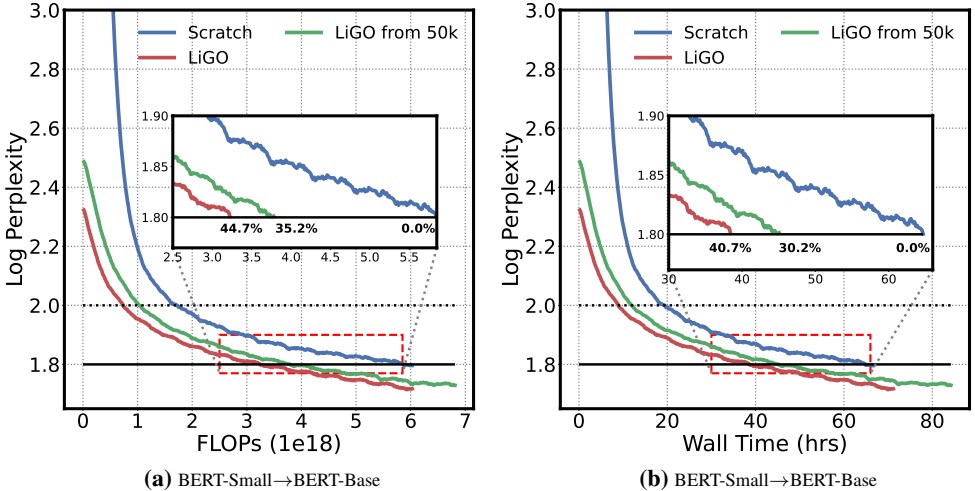

**(a)** BERT-Small→BERT-Base      **(b)** BERT-Small→BERT-Base

**Figure 7:** Results on BERT-Base by reusing BERT-Small trained for 50k steps. Instead of training BERT-Base from fully trained BERT-Small, we run LiGO on BERT-Small trained with 50k steps. LiGO offers about 35.2% savings in FLOPs and 30.2% savings in wall time over the BERT-Base training from scratch.

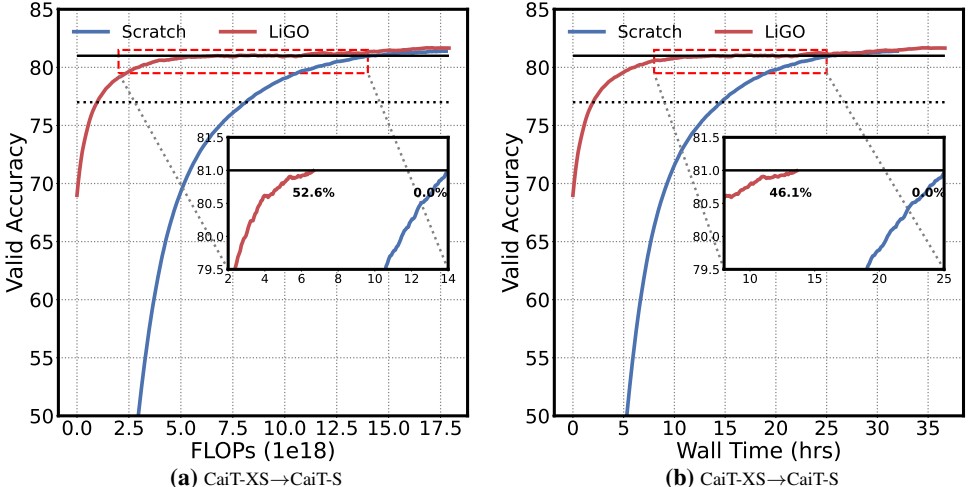

**(a)** CaiT-XS→CaiT-S      **(b)** CaiT-XS→CaiT-S

**Figure 8:** Results on CaiT. (a) Accuracy vs. flops and (b) accuracy vs. wall time for training CaiT-S. LiGO saves flops by 52.6% and wall time by 46.1% over training from scratch on ImageNet.

**Table 5:** GLUE performance of different LiGO models. All of the results are based on BERT-Base models with BERT-Small as the base model for LiGO optimization.

| Method | SST-2 (Acc.) | MNLI (Acc.) | MRPC (Acc.) | CoLA (Acc.) | QNLI (Acc.) | QQP (Acc.) | STS-B (Acc.) | Average (Acc.) |
|---|---|---|---|---|---|---|---|---|
| BERT-Small (Scratch) | 87.21 | 77.56 | 82.11 | 59.93 | 85.06 | 85.82 | 84.99 | 80.38 |
| BERT-Base (LiGO Init) | 88.15 | 77.62 | 82.53 | 60.70 | 85.79 | 86.65 | 85.83 | 81.04 |
| BERT-Base (LiGO Init + Pretrain) | 88.42 | 79.29 | 84.31 | 62.09 | 88.07 | 88.81 | 87.00 | 82.57 |
| BERT-Base (Scratch) | 88.19 | 78.43 | 85.78 | 62.09 | 87.06 | 87.18 | 86.99 | 82.25 |

**Table 6:** Downstream performance using AdapterFusion (Pfeiffer et al., 2020) on GLUE Benchmark. All of the results are based on BERT-Base models trained using different baselines.

| Method | Savings (FLOPs) | Savings (Walltime) | SST-2 (Acc.) | MNLI (Acc.) | MRPC (Acc.) | CoLA (Acc.) | QNLI (Acc.) | QQP (Acc.) | STS-B (Acc.) | Average (Acc.) |
|---|---|---|---|---|---|---|---|---|---|---|
| Scratch | – | – | 88.41 | 78.60 | 86.02 | 62.39 | 87.62 | 88.02 | 86.52 | 82.51 |
| StackBERT | 34.1% | 33.3% | 88.78 | 79.80 | 85.43 | 59.56 | 87.71 | 89.19 | 86.27 | 82.39 |
| MSLT | 34.9% | 30.0% | 88.41 | 78.35 | 83.15 | 63.97 | 86.19 | 88.20 | 86.42 | 82.10 |
| KI | -5.7% | -13.9% | 88.94 | 78.84 | 84.00 | 64.61 | 86.75 | 88.19 | 87.93 | 82.75 |
| bert2BERT | 29.0% | 25.1% | 88.47 | 80.53 | 85.50 | 62.33 | 88.57 | 86.72 | 87.10 | 82.75 |
| LiGO | 44.7% | 40.5% | 88.45 | 80.01 | 84.67 | 63.05 | 88.06 | 88.92 | 87.00 | 82.88 |

on-par performance with model trained from scratch under adapter-based tuning with 44.7% savings in FLOPs abd 40.5% savings in wall time. This shows that LiGO does not harm the model generalization capability when adapters are used as a parameter-efficient finetuning strategy for transferring a trained model to downstream datasets.

## C.5 INITIAL RESULTS ON BILLION+ PARAMETER MODELS

Our extensive experiments on BERT (Devlin et al., 2019), RoBERTa (Liu et al., 2019), GPT2 (Radford et al., 2019), DeiT (Touvron et al., 2021a) and CaiT (Touvron et al., 2021b) show that LiGO can consistently improve transformer training efficiency over the traditional way of training from scratch across domains and model sizes. One interesting future direction of our work is scaling LiGO to very large models with parameters more than 100B, such as GPT3 (Brown et al., 2020). While we currently do not possess the compute resources for this extreme large-scale study, we perform a preliminary experiment on GPT2-1.5B (Radford et al., 2019) by using GPT2-Medium as the initialization. We train for 15k steps on C4 dataset (Raffel et al., 2020) and find that our proposed LiGO saves about 39% computation cost (FLOPs) of training GPT2-1.5B from scratch to reach the same log perplexity (which is 3.3). We believe that it is imperative to study the extent to which the benefits of LiGO remain at the scale on which the modern large language models are trained. We hope to cover this in our future work.

