# OpenReview forum: "Learning to Grow Pretrained Models for Efficient Transformer Training"
_ICLR.cc/2023/Conference — ICLR 2023 notable top 25%_

### Official Review · Reviewer_bupJ · 2022-10-24

**Confidence:** 4
**Correctness:** 4
**Technical Novelty And Significance:** 3
**Empirical Novelty And Significance:** 3
**Recommendation:** 8

**Clarity, Quality, Novelty And Reproducibility:**

The paper is written clearly and the work is novel. Please refer to the strengths and weaknesses section above for details.

**Details Of Ethics Concerns:**

None.

**Strength And Weaknesses:**

**Strengths**

1. The paper is written well and easy to follow.
2. Given the prevalence of large (foundation) models, and the fact that compute is perhaps the most important constraint while training these models, the paper tackles a very good problem in a timely manner.
3. The experiments aptly make the points of the main paper.

**Weaknesses/ Questions/ Suggestions**

1. **Performance of the base model**: I would encourage the authors to report the performance of the base (small) models, from where the larger models are trained, in the plots. It would help us clarify a) how quickly the larger model recovers the performance of the base model, b) how much it improves upon the base model.

2. I would encourage the authors to modify the introduction/ story and emphasize the fact that their proposed method allows one to train a family of Transformer models, where new and larger models can utilize the information in the previous and smaller models to save compute, which is not possible when training from scratch. In the absence of this messaging, one could fairly ask, when comparing model growth with training from scratch, the compute budget of training the smaller models should also be included in the plots.

3. **Scalability to larger models**: The authors have reported performance on relatively smaller architectures (up to 774 million parameters) compared to modern Transformer models. I wonder if the proposed method would also result in equivalent gains on these larger models. I see that in the conclusion the authors already concede this point but, nevertheless, I would like to know the authors' take on that.


**Summary Of The Paper:**

The paper presents a learning strategy to grow smaller pre-trained networks into larger networks with the hope that model growth can save compute compared to a network that is trained from scratch. While mostly in the literature, growth strategies are based on heuristics, e.g., copying the layers of the network for expanding depth, or block diagonal expansion for increasing width, learning-based model growth is not extensively studied. In this work, the authors initialize the parameters of the larger network as a linear transformation of the parameters of the smaller network. This linear transformation is learned through data using a small number of gradient steps (typically ~100) – a cost which is marginal compared to the training of the larger network. The authors propose to decompose the linear transformation into depth- and width-operators, and use structured sparse matrices for these operators. For enforcing structures and sparsity, layer-wise transformations are proposed using kronecker factorization. Through experiments on BERT, RoBERTA, GPT2 and ViT, the authors show that their proposed growth strategy saves up to ~40% of compute without sacrificing the upstream and downstream performance of Transformer models across the domains.

**Summary Of The Review:**

Please refer to the strengths and weaknesses section above for details.

---

> ### Author Response · Authors · 2022-11-17
> **Response to Reviewer bupJ**
>
> We thank Reviewer bupJ for the positive recommendation and constructive comments.
>
> (a) **Performance of the base (smaller) transformer:** Thanks for mentioning this. To indicate the performance of base/small models, we have added another thinner dotted horizontal line in our figures (see Figure 2, Figure 3 and Figure 4 in the revised version). We find LEGO uses only $0.8 \times 10^{18}$ FLOPs on BERT, $0.5 \times 10^{18}$ FLOPs on RoBERTa, and $1.7 \times 10^{18}$ FLOPs on DeiT, to reach the smaller model performance.
>
> (b) **Modify introduction to emphasize availability of smaller models:** Thanks for this great suggestion. As rightly pointed by reviewer, LEGO assumes that there exists a smaller training model, and thus focuses on utilizing the knowledge of smaller models that have already been pretrained and available. We have changed the introduction to emphasize this point in our revised paper.
>
> \(c\) **Scalability to very large models:** One interesting future direction of our work is scaling LEGO to very large models with parameters more than 100B, such as GPT3. While we currently do not possess the compute resources for this extreme large-scale study, during the rebuttal phase we performed a preliminary experiment on GPT2-1.5B by using GPT2-Large as the initialization. We trained for 15k steps on C4 dataset (during the rebuttal time window) and find that LEGO can save about 39% computation cost of training GPT2-1.5B from scratch to reach the same log perplexity (which is 3.3). We agree with the reviewer and believe that it is imperative to study the extent to which the benefits of LEGO remain at the scale on which the modern large language models are trained. We hope to cover this in our future work. We have added this discussion in Appendix J of the revised draft.

---

> > ### Comment · Reviewer_bupJ · 2022-11-22
> > **Rebuttal acknowledgment**
> >
> > I read the authors' rebuttal. I am satisfied with their response and, therefore, keep my score.

---

### Official Review · Reviewer_xesb · 2022-10-25

**Confidence:** 3
**Correctness:** 3
**Technical Novelty And Significance:** 3
**Empirical Novelty And Significance:** 3
**Recommendation:** 8

**Clarity, Quality, Novelty And Reproducibility:**

Simple but it’s a good try. Overall this paper is of good quality, with detailed formulas and clear diagrams, and a certain degree of originality.

**Strength And Weaknesses:**

Strength
1.This paper uses a subtle factorization which substantially reduces the number of computational parameters, making it feasible to learn such a linearly map. This idea is very good, simple but useful, and also has some explainability.
2.Clearly illustrates how to extend the parameters in depth and width from a theoretical point of view, and with some justification.
3.The paper is well written.
Weakness
1.The training process of LEGO method is not described in detail in the paper, although illustrated in the appendix, I suppose it should be described in detail in the main paper in Sec.3 or another place. And the optimization process is not mentioned either. As I understand it, the parameters of LEGO are trained for a specific number of steps together with the parameters of the larger model, and then the training of LEGO is stopped to train only the larger model. Is it right?
2. Compared to training from sketch, I wonder how about the performance in downstream task of model which use LEGO method initialized. The performance of downstream task is what we all concern, and it’s important. I hope you can add some relevant experiments.
3.Only one Vision Transformer model has experimented, more vision models should be considered.


**Summary Of The Paper:**

The paper proposes a method for initializing a larger model by learning a linearly map using the parameters of a small model to obtain the parameters of a larger one, thus reducing the training cost of the larger model by using the existing small model.
Kronecker factorization is used to reduce the computational parameters to an acceptable level, making it possible to learn a linearly map as mentioned above.
The proposed method is applicable not only to multilayer neural network structures but also to transformers, and experiments have been conducted on various transformer-based models with good results.

**Summary Of The Review:**

The idea and method in this paper is novel and subtle, makes it easy to learn a linearly map to grow model. However, the specific training process of LEGO is not highlighted in the text, and the experimental part is not sufficient, such as the detailed evaluation of the performance of downstream tasks.

---

> ### Author Response · Authors · 2022-11-17
> **Response to Reviewer xesb**
>
> We thank Reviewer xesb for appreciating our core idea and theoretical perspective. We have addressed all the questions and experiments as described below, with corresponding changes in the revised version.
>
> (a) **Training and optimization details:** At the beginning, we fix the parameters of small models and only tune the coefficients in the LEGO operator for few steps (100 steps in our experiments). Since we only tune for a 100 stops, the FLOPS/compute coming from the LEGO operator is negligible compared to the training cost (however we include this cost in all our tables and figures).  Afterwards, we leverage the learned LEGO to grow a small model to a large model. Then we proceed regular training with the grown model. We have included a summary of overall LEGO training in our revised paper.
>
> (b) **Downstream task performance with LEGO initialization:** This is an extremely interesting suggestion.
> Following reviewer's suggestion, we performed an experiment by directly finetuning a BERT-Base model initialized by LEGO (from BERT-Small) without any pre-training and observe that LEGO initialized model can benefit downstream tasks compared to a BERT-Small model trained from scratch (1st row vs 2nd row). This suggests that we can simply expand with the LEGO operator for a 100 steps and then perform downstream finetuning directly to improve performance over finetuning the smaller model. RTable 1 summarizes the results. We have added this additional result in Appendix I of the revised draft.
>
> RTable 1: GLUE performance of different LEGO models. All of the results are based on BERT-Base models with BERT-Small as the base model for LEGO optimization.
> | Model                       |   SST-2 |   MNLI |   MRPC |   RTE |   QNLI |   QQP |   STS-B |   Avg. GLUE |
> |:----------------------------|--------:|-------:|-------:|------:|-------:|------:|--------:|------------:|
> | BERT-Small (Scratch)        |   87.21 |  77.56 |  82.11 | 59.93 |  85.06 | 85.82 |   84.99 |       80.38 |
> | BERT-Base (LEGO Init)       |   88.15 |  77.62 |  82.53 | 60.70 |  85.79 | 86.65 |   85.83 |       81.04 |
> | BERT-Base (LEGO + Pretrain) |   88.42 |  79.29 |  84.31 | 62.09 |  88.07 | 88.81 |   87.00 |       82.57 |
> | BERT-Base (Scratch)         |   88.19 |  78.43 |  85.78 | 62.09 |  87.06 | 87.18 |   86.99 |       82.25 |
>
>
> \(c\) **Experiments on another vision transformer:** Thanks for the suggestion! In addition to DeiT, we perform additional experiments with CaiT [1] on ImageNet and find that while reusing CaiT-XS, LEGO offers about 52.6% savings in FLOPs and 46.1% savings in wall time over the CaiT-S training from scratch. We have added this result in the updated version (see Figure 8 in Appendix H).
>
> **References:**
>
> [1] Hugo Touvron, Matthieu Cord, Alexandre Sablayrolles, Gabriel Synnaeve, Hervé Jégou. Going deeper with Image Transformers. ICCV, 2021.

---

### Official Review · Reviewer_nRWA · 2022-10-25

**Confidence:** 3
**Clarity, Quality, Novelty And Reproducibility:** This paper is well presented, but the…
**Correctness:** 4
**Technical Novelty And Significance:** 3
**Empirical Novelty And Significance:** 2
**Recommendation:** 6

**Strength And Weaknesses:**

Strengths:
1. The idea of this paper is easy to follow. By learning the expansion operators with task loss, the pretrained smaller transformer can be naturally mapped to a larger network, which could obtain a significantly higher accuracy at the beginning of training compared to training from scratch.

2. Extensive experiments on various tasks validate the efficacy of the proposed method. For example, in Table 1, LEGO outperforms Scratch in most cases with 40.5% training time reduced.

Weaknesses:
1. The method requires a trained smaller transformer to initialize the target network, but I'm not sure whether the training cost is still low if we take the training time of the smaller network into account.
2. It seems that LEGO only supports expanding the width of a linear transformation or growing it into a sequence of linear transformations, while two transformations connected by a non-linear activation function can not be initialized by one transformation. This may be a limitation of this work.
3. The improvements compared to existing methods such as StackBERT are marginal. In Figure 2, StackBert has lower log perplexity curves when FLOPs > 4e18 or wall time > 40 hrs.


**Summary Of The Paper:**

This paper proposes a method named LEGO to efficiently train a transformer model by initializing it with a pretrained smaller transformer. The method expands the pretrained transformer model on width and depth by learning linear maps on the parameters. Experiments on various language and vision tasks show that LEGO can obviously reduce the training time compared to training from scratch.

**Summary Of The Review:**

Overall, I think this paper is a borderline paper. My major concerns of this paper are the limited improvements compared to the existing methods and the generalizability of the method.

---

> ### Author Response · Authors · 2022-11-17
> **Response to Reviewer nRWA**
>
> We thank the reviewer for the thoughtful reviews and great suggestions. Below are our responses to the concerns and we have incorporated all the feedback in the revised version.
>
> (a) **Training time of smaller transformer:** Similar to prior works like bert2BERT and KI, LEGO assumes that there exists a smaller training model, and thus focuses on utilizing the knowledge of smaller models that have already been pretrained and available. In all our experiments, the smaller models follow the standard model configurations. One can easily acquire such model weights from some public model zoos, e.g., Huggingface. We have changed the introduction to emphasize this point in the revised paper, as suggested by Reviewer bupJ.
>
> However, we think the reviewer's suggestion is an interesting one (i.e., can LEGO enable faster training of models from scratch by training a smaller model first and then growing it during training), and thus investigated this setting in more detail. We compute the training FLOPs and walltime of BERT-Small (6L-512H) for 220k steps, which are $1.10 \times 10^{18}$ FLOPs and 18 hours, respectively. Compared with the total FLOPs of training a larger model, these costs are relatively much cheaper. Note that our LEGO can also leverage smaller existing models that are only trained for few steps to accelerate training of a larger model. To verify this, we perform an experiment by reusing a BERT-Small trained for 50k steps and find that LEGO can still save 35.2% savings in FLOPs and 30.2% savings in wall time over the BERT-Base training from scratch. Thus, LEGO can enable faster training of larger models even if the smaller model has to be trained from scratch. We have added this additional result in Figure 7 of Appendix F in the revised paper.
>
> (b) **LEGO with two transformations connected by a non-linear activation function:** We may be misunderstanding, but LEGO is a generic method and can in fact handle two linear layers connected by a non-linear activation function (e.g., ReLU). Note that different from Net2Net which requires the grown model to preserve the functionality, our growth strategy focuses on only minimizing the objective mentioned in Eq. 4 of the revised draft. Thus, there is no need to handle non-linearity particularly. Empirically, we applied LEGO to the FFN block in transformer, and no degradation has been observed during the training.
>
> \(c\) **Improvements over existing methods/StackBERT:** We do not think our improvements are marginal compared with all the baselines. Among BERT, RoBERTa, and ViT, LEGO outperforms all the baselines by about ~10% in both FLOPs and walltime, demonstrating its efficacy over existing methods in accelerating training of transformers. When compared to StackBERT, LEGO obtains +10.6% (34.1 vs 44.7) and +31.6% (23.8 vs 55.4) improvements in FLOPs for BERT and DeiT respectively.
>
> In Figure 2, although StackBERT achieves lower log perplexity when FLOPs > 4e$^{18}$, both LEGO and StackBERT leads to very similar log perplexity (1.71 vs 1.72) at the end of training. We also check the downstream GLUE performance of both models at the end of training and find that they obtain very similar performance, as shown in RTable 1. We have added this additional result in Appendix G of the revised draft.
>
> RTable 1: GLUE performance of StackBERT and LEGO at the end of training. All of the results are based on BERT-Base trained using BERT-Small as the smaller model.
> | Model     |   SST-2 |   MNLI |   MRPC |   RTE |   QNLI |   QQP |   STS-B |   Avg. GLUE |
> |:----------|--------:|-------:|-------:|------:|-------:|------:|--------:|------------:|
> | StackBERT |   88.57 |  79.84 |  85.06 | 61.21 |  88.25 | 89.49 |   87.02 |       82.78 |
> | LEGO      |   88.55 |  79.89 |  84.98 | 61.54 |  88.60 | 88.63 |   87.14 |       82.76 |

---

### Official Review · Reviewer_52nq · 2022-10-30

**Confidence:** 3
**Correctness:** 4
**Technical Novelty And Significance:** 3
**Empirical Novelty And Significance:** Not applicable
**Recommendation:** 8

**Clarity, Quality, Novelty And Reproducibility:**

Overall, the paper is well-written and easy to follow. The novelty of the method seems incremental, but the empirical results are good. Sufficient details (e.g., experimental setup) are described in the paper such that an expert should be able to reproduce the main results

**Strength And Weaknesses:**

Pros:
* The first work to combine both width- and depth-growth operators together and thus expand the model in both width and depth.
* LEPO shows good capacity in both the linear layer and the attention layer.
* The empirical results show that LEPO can greatly improve training efficiency.

Questions:
* The reported FLOPs and Wall Time of LEPO are counted during training. Would it be more reasonable to also consider the calculation of the LEGO operator before the transformer training?
* As claimed in the experiment part, the authors claim that they investigate the performance of LEGO when transferring the model to other downstream tasks. Do the authors compare LEGO with some adapter-based methods or visual prompt tuning methods?
* Some typos can be fixed. For example, the reported results of LEGO show it can reduce 40.5% wall time when transferring BERT-Small to BERT-Base, however in Figure 2(b) the number is 40.7%.

**Summary Of The Paper:**

This paper investigates the problem of accelerating the large-scale transformer model's training. The idea is to initialize the large-scale model with the transformed parameters of the small-scale model. Compared to existing methods, the proposed method can expand the network in both width and depth and the Kronecker factorization is used to reduce the computational cost. The experimental results in both BERT and Vision Transformer demonstrate that LEGO can make the model converge faster.

**Summary Of The Review:**

Please refer to the 'Clarity, Quality, Novelty And Reproducibility' part above.

---

> ### Author Response · Authors · 2022-11-17
> **Response to Reviewer 52nq**
>
> We thank Reviewer 52nq for acknowledging that ours is the first work to combine both width- and depth-growth operators together and thus expand the model in both width and depth. Below are our responses on the additional clarification and experiments.
>
> (a) **FLOPs and wall time of LEGO before the transformer training:** Thanks for pointing this out. In fact, we already include the FLOPs for training a LEGO operator in both our tables and figures. However, since our LEGO operator is parameter efficient and we only train it for 100 steps, the contribution of the LEGO operator to the total FLOPS/compute time is negligible. For example, while reusing BERT-Small, training the LEGO operator takes only $3.61 \times 10^{15}$ FLOPs, which is negligible compared with total $7 \times 10^{18}$ FLOPs of BERT-Base (0.00051% of total compute). We have claried this in our revision.
>
> (b) **LEGO with adapter-based methods:** Thanks for the interesting suggestion. Our LEGO is mainly proposed for improving efficiency of the pre-training stage and hence is compatible with various fine-tuning schemes like full model finetuning, adapters or prompt tuning for adaptation to downstream tasks. For full model finetuning, Table 1 and Table 2 show that despite being trained significantly faster than scratch, BERT and DeiT trained with our LEGO approach both achieve very similar performance comparing to the baselines on downstream tasks. Following reviewer's suggestion, we also tested BERT-Base
> models trained using different baselines by using adapters [1] instead of full finetuning on GLUE benchmark. The results are listed as below:
>
> RTable 1: Downstream Performance using AdapterFusion [1] on GLUE Benchmark. All of the results are based on BERT-Base models trained using different baselines.
>
> | Model     |   Saving FLOPs | Saving Time | SST-2 |   MNLI |   MRPC |   RTE |   QNLI |   QQP |   STS-B |   Avg. GLUE |
> |:----------|--------:|--------:|--------:|-------:|-------:|------:|-------:|------:|--------:|------------:|
> | Scratch   | -- | -- |   88.41 |  78.60 |  86.02 | 62.39 |  87.62 | 88.02 |   86.52 |       82.51 |
> | StackBert | 34.1% | 33.3% |   88.78 |  79.80 |  85.43 | 59.56 |  87.71 | 89.19 |   86.27 |       82.39 |
> | MSLT      | 34.9% | 30.0% |   88.41 |  78.35 |  83.15 | 63.97 |  86.19 | 88.20 |   86.42 |       82.10 |
> | KI        | -5.7% | -13.9% |   88.94 |  78.84 |  84.00 | 64.61 |  86.75 | 88.19 |   87.93 |       82.75 |
> | bert2BERT | 29.0% | 25.1% |   88.47 |  80.53 |  85.50 | 62.33 |  88.57 | 86.72 |   87.10 |       82.75 |
> | LEGO      | 44.7% | 40.5% |   88.45 |  80.01 |  84.67 | 63.05 |  88.06 | 88.92 |   87.00 |       82.88 |
>
> Our new experiment implies LEGO also achieves on-par performance with model trained from scratch under adapter-based tuning with 44.7% savings in FLOPs abd 40.5% savings in wall time. We have added these results in Appendix E of the revised draft.
>
> \(c\) **Typos:** Thanks for pointing them out. We have fixed all the typos in our revision.
>
> **References:**
>
> [1] Jonas Pfeiffer, Aishwarya Kamath, Andreas Rücklé, Kyunghyun Cho, Iryna Gurevych. AdapterFusion: Non-Destructive Task Composition for Transfer Learning. EACL, 2021.

---

> > ### Comment · Reviewer_52nq · 2022-11-24
> > **Rebuttal acknowledgement**
> >
> > The authors have well addressed my concerns. As a result, I am raising my score from 6 to 8.

---

### Author Response · Authors · 2022-11-17
**Summary of Author's Response**

We would like to thank all the reviewers for their constructive comments! We are glad that the reviewers found that: (a) our paper tackles a very good problem in a timely manner given the prevalence of large (foundation) models (R-bupJ); (b) ours is the first work to combine both width- and depth-growth operators together (R-52nq) from theoretical point of view (R-xesb); \(c\) our idea is very good, simple (R-xesb), and novel (R-xesb, R-bupJ); (d) our extensive experiments on various tasks aptly make the points of the main paper (R-bupJ), by showing great improvements in training efficiency (R-52nq, R-nRWA), including good capacity in both the linear layer and the attention layer (R-52nq).

We have addressed all the questions that the reviewers posed with additional experimental comparisons and clarifications. All of these additional experiments and suggestions have been added into the updated PDF. Below, we summarize the main changes to the paper and request the reviewers to take a look at the new additions.

- Performance of LEGO with adapter-based methods, as suggested by R-52nq,
- Clarification and additional experiment related to training time of smaller transformer, as suggested by R-nRWA,
- Downstream task performance with LEGO initialization, as suggested by R-xesb,
- Experiments on another vision transformer, as suggested by R-xesb,
- Added performance of the base (smaller) transformers in figures, as suggested by R-bupJ,
- Discussion on scalability to very large models, as suggested by R-bupJ.

---

### Decision · Program_Chairs · 2023-01-20

**Decision:**

Accept: notable-top-25%

**Justification For Why Not Higher Score:**

The overall idea of this paper is interesting. But actually, the idea could be heuristic and natural to improve the training of large-scale models. Importantly, the proposed algorithm needs the parameters of small-scale models as the 'warming up', which somehow is a practical constraint. A spotlight presentation would be good already to give this paper enough exposure to the relevant research community or audience.

**Justification For Why Not Lower Score:**

Almost all reviewers think this paper is well-written and easy to follow. The authors focus on the interesting problem addressing the constraints during the training of large-scale Transformer models. The novelty of the method seems incremental, but the empirical results are good. The authors also successfully evaluated the model's to very large models during the rebuttal time window, which is very nice.

**Metareview: Summary, Strengths And Weaknesses:**

This paper is to accelerate the training of large-scale Transformer models by initializing their parameters with the parameters of the small-scale models. Also the authors expand the pretrained transformer model's width and depth by learning linear maps of the parameters. Extensive experiments have been done to support the effectiveness of the proposed algorithm.

**Note From Pc:**

if the above contains the word "oral" or "spotlight" please see: "oral" presentation means -> notable-top-5% and "spotlight" means -> notable-top-25%. As stated in our emails, we are disassociating presentation type from AC recommendations